Resource
# Conservation of cell-intrinsic immune responses in diverse nonhuman primate species

Jenna M Gaska[1], Lance Parsons[2] ⓘ, Metodi Balev[1], Ann Cirincione[1], Wei Wang[2], Robert E Schwartz[3], Alexander Ploss[1] ⓘ

**Differences in immune responses across species can contribute to the varying permissivity of species to the same viral pathogen. Understanding how our closest evolutionary relatives, nonhuman primates (NHPs), confront pathogens and how these responses have evolved over time could shed light on host range barriers, especially for zoonotic infections. Here, we analyzed cell-intrinsic immunity of primary cells from the broadest panel of NHP species interrogated to date, including humans, great apes, and Old and New World monkeys. Our analysis of their transcriptomes after poly(I:C) transfection revealed conservation in the functional consequences of their response. In mapping reads to either the human or the species-specific genomes, we observed that with the current state of NHP annotations, the percent of reads assigned to a genetic feature was largely similar regardless of the method. Together, these data provide a baseline for the cell-intrinsic responses elicited by a potent immune stimulus across multiple NHP donors, including endangered species, and serve as a resource for refining and furthering the existing annotations of NHP genomes.**

## Introduction

Many of the pathogens with a significant impact on human morbidity and mortality—such as HIV, hepatitis C virus, hepatitis B virus, and yellow fever virus—have a narrow host range limited to humans and select nonhuman primate (NHP) species ([1]). Broadly speaking, the host tropism of such viral pathogens can be determined by (i) the absence or incompatibility of (a) host factor(s) needed for part(s) of the viral life cycle; (ii) the presence of dominant restriction factors; and/or (iii) differences in immune responses. Understanding the molecular basis of host tropism can inform our understanding of intra- and interspecies transmission and aid in the generation of improved animal models and clinical therapies. Here, we analyzed how cell-intrinsic immune responses compare across humans and NHP species. As our closest evolutionary relatives,

the tactics by which NHPs have evolved to confront pathogens and the corresponding ways those pathogens modulate host immune defenses—as highlighted by hotspots for positive selection in genes involved with immune/defense responses in humans and select NHPs (([2], [3], [4], [5]), reviewed in reference [6])—could shed light on host range barriers pertinent to studying zoonotic infections and using NHPs as biomedical research models.

In the present study, we purposefully chose a cell type that could be easily collected from a wide array of multiple NHP species, was conducive to studies of cell-intrinsic immune responses and could be used for future generation of induced pluripotent stem cells to study host–pathogen interactions in desired cell lineages. The latter criterion would be especially advantageous for studying pathogens with both highly limited host and tissue tropism, such as hepatitis C virus, for which acquiring hepatocytes from NHPs can be extremely challenging for both ethical and financial reasons. Dermal fibroblasts (DFs) met all these requirements and could be obtained through biopsies or existing repositories. As an added advantage, we used only primary DFs which do not have the cell cycle dysregulation or disruption of immune signaling as observed for immortalized cells such as human hepatoma cell lines ([7], [8]).

We selected three donors from eight different NHP species covering ~35 million years of evolution (Fig 1A and Table 1): great apes (*Pan troglodytes, Pan paniscus, Gorilla gorilla,* and *Pongo pygmaeus*), because of their close genetic relationship to humans but whose endangered status precludes their use in most scientific studies; three Old World monkey species (*Papio anubis, Macaca nemestrina,* and *Macaca mulatta*); and one New World species (*Saimiri sciureus*), which are all commonly used in biomedical research. As a more distant point in evolutionary time (ca. 65 million years since divergence), we also included mouse. For our initial studies comparing cell-intrinsic immunity in these DFs from diverse species, we aimed to provoke an immune response independent of a given species' susceptibility and/or permissivity to a virus. Thus, we used polyinosinic:polycytidylic acid (poly(I:C)), a synthetic dsRNA analog that can be transfected into cells and potently stimulates an IFN-mediated response by resembling a pathogen-associated molecular pattern characteristic of RNA virus infections (Fig 1B) ([9], [10], [11]). This allowed us to compare the repertoire of cell-intrinsic

[1]Lewis Thomas Laboratory, Department of Molecular Biology, Princeton University, Princeton, (NJ, USA    [2]Carl Icahn Laboratory, Lewis-Sigler Institute for Integrative Genomics, Princeton University, Princeton, NJ, USA    [3]Weill Cornell Medical College, Belfer Research Building, New York, NY, USA

Correspondence: aploss@princeton.edu

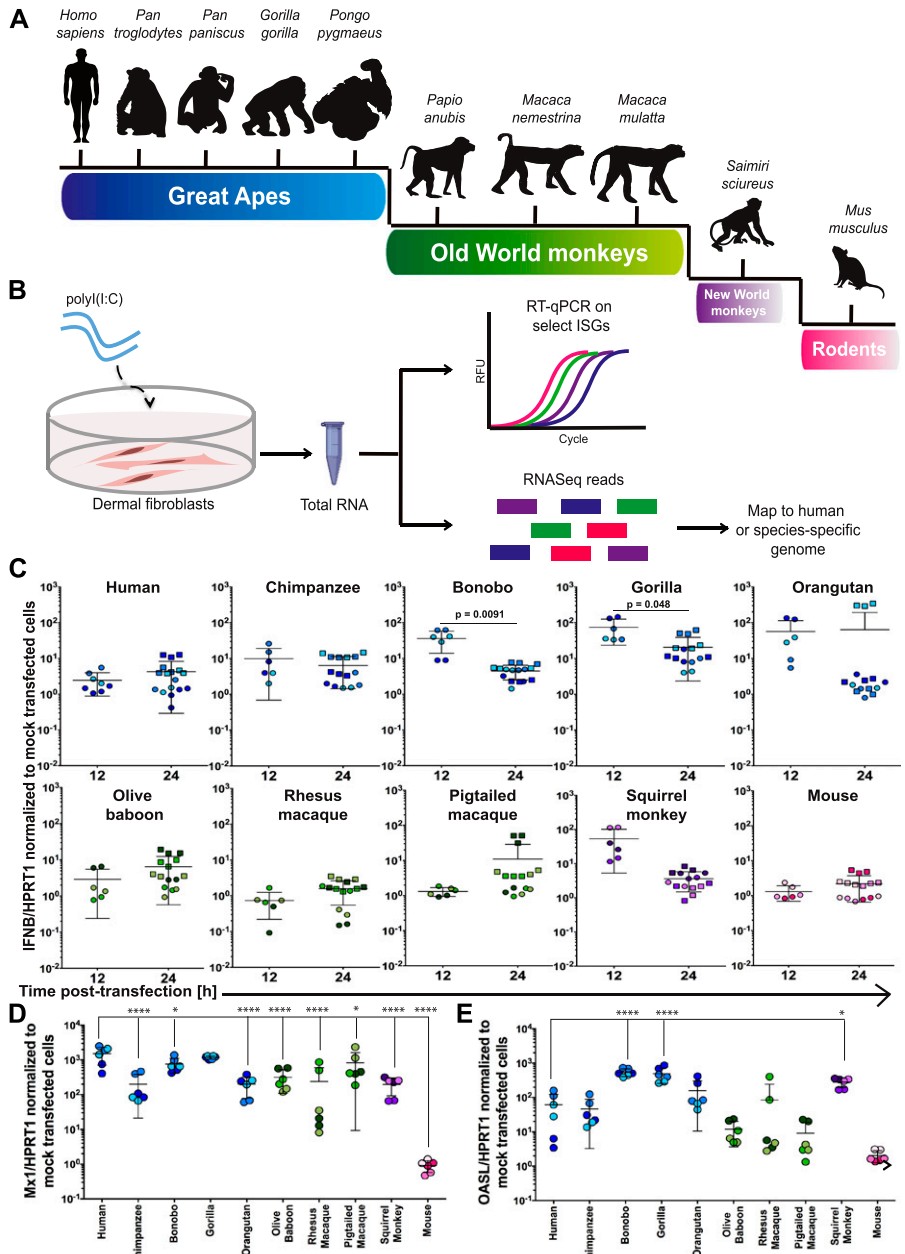

**Figure 1. Comparing the cell-intrinsic immune response to poly(I:C) across multiple species.**
**(A)** Schematic representation of the phylogenetic relationship of species included in this study. **(B)** Overview of the experimental workflow. **(C, D, E)** Using RT-qPCR, *IFNβ* (C), *MX1* (D), and *OASL* (E) mRNA expression were assessed in primate and mouse primary DFs relative to the housekeeping gene *HPRT1* at 12 and/or 24 h after poly(I:C) (~53 ng/cm²) or mock transfection. Values shown are the fold changes relative to mock-transfected cells. Each data point represents an individual well of cells and is colored by donor (see Fig S2 for the color code used). Circles represent data from 24-well experiments, whereas squares are from the transfections performed in six-well plates that underwent RNA-Seq. Data information: In (C, D, E), bars depict the mean with SD. Note that the lower error bar for the orangutan samples at both time points and the pig-tailed macaque at 24 h are missing because they approach a value of 0. Where $P ≤ 0.05$, the $P$ value is indicated. In (C), an unpaired $t$ test was used to compare timepoints, with Welch's correction performed where the SD between time points differed by more than a factor of 2. In (D, E), an ordinary one-way ANOVA was performed followed by Dunnett's multiple comparisons test with a single pooled variance and the human samples serving as the control against which all comparisons were made. *$P ≤ 0.05$; **$P ≤ 0.01$; ***$P ≤ 0.001$; ****$P ≤ 0.0001$. Source data are available for this figure.

responses available to DFs from each of these species unhindered by pathogen-mediated modulation that can occur as early as viral binding and entry.

We subsequently analyzed the transcriptomic profile of these cells, generating data in response to a mimic of viral infection in the broadest panel of NHP species compiled to date. Although the evolution of select factors in response to host–pathogen conflict has been explored (12, 13), there has not been a comprehensive analysis of cell-intrinsic responses in nonimmune cells across a diverse range of NHPs. Much of comparative primate immunology focuses on the major cellular constituents of innate and adaptive immunity (14, 15, 16, 17, 18, 19) and highly variable genetic loci such as those encoding MHC class I molecules (20, 21, 22), immunoglobulins

(23), and MHC class I–related (MR1) molecules (24). Furthermore, such studies often use a limited number of NHP species and focus on differences at the genetic level versus expression. To our knowledge, the closest study to ours examining the functional consequences of differences and similarities in innate immunity in NHPs stimulated primary monocytes from chimpanzee, rhesus, and human with LPS (25). Two more recent studies also provided insights into cell-intrinsic immunity but on a much larger evolutionary scale, including human or human and rhesus macaque as the only primates (26, 27).

The immense transcriptomic dataset we have acquired and analyzed in the course of this work fulfills a need for not only better understanding cell-intrinsic immunity in our closest evolutionary

**Table 1.** Known information for each DF donor.

|  | Common name | Species name | Classification | ID | Biopsy site | Sex | Age | Passage frozen |
|---|---|---|---|---|---|---|---|---|
| Donor 1 | Pig-tailed macaque | M. nemestrina | Old World Monkey | AG07923 | Skin, arm | Male | 15 yr | 2 |
| Donor 2 | Pig-tailed macaque | M. nemestrina | Old World Monkey | AG08490 | Skin, arm | Male | 22 yr | 2 |
| Donor 3 | Pig-tailed macaque | M. nemestrina | Old World Monkey | PR00058 | Skin, skin | Female | 2 yr | 2 |
| Donor 1 | Olive baboon | P. anubis | Old World Monkey | PR00033 | Skin, skin | Male | 3 yr | 1 |
| Donor 2 | Olive baboon | P. anubis | Old World Monkey | PR00039 | Skin, skin | Male | 3 yr | 2 |
| Donor 3 | Olive baboon | P. anubis | Old World Monkey | PR00036 | Skin, skin | Male | 3 YR | 1 |
| Donor 1 | Squirrel monkey | S. sciureus | New World Monkey | AG05311A | Skin, skin | Female | UNKNOWN | 14 |
| Donor 2 | Squirrel monkey | S. sciureus | New World Monkey | SQMA | Skin, stomach | Male | 3 yr | 4 |
| Donor 3 | Squirrel monkey | S. sciureus | New World Monkey | SQMB | Skin, stomach | Male | 3 yr | 4 |
| Donor 1 | Orangutan | P. pygmaeus | Great ape | AG06105 | Skin, thoracolumbar junction | Female | 26 yr | 5 |
| Donor 2 | Orangutan | P. pygmaeus | Fibroblast | PR00054 | Skin, skin | Male | 4 yr | 5 |
| Donor 3 | Orangutan | P. pygmaeus | Great ape | PR01109 | Skin, leg | Female | 40 yr | 2 |
| Donor 1 | Gorilla | G. gorilla | Great ape | PR00230 | Skin, skin | Female | 37 yr | 2 |
| Donor 2 | Gorilla | G. gorilla | Great ape | PR00573 | Skin, inner thigh | Male | 34 yr | 2 |
| Donor 3 | Gorilla | G. gorilla | Great ape | PR00107 | Skin, skin | Male | 19 yr | 3 |
| Donor 1 | Bonobo | P. paniscus | Great ape | PR00111 | Skin, location unknown | Male | 31 yr | 4 |
| Donor 2 | Bonobo | P. paniscus | Great ape | PR00235 | Skin, skin | Female | 28 yr | 5 |
| Donor 3 | Bonobo | P. paniscus | Great ape | PR00248 | Skin, skin | Female | 1 yr | 2 |
| Donor 1 | Chimpanzee | P. troglodytes | Great ape | S004933 | Skin, location unknown | Female | 6 yr | 2 |
| Donor 2 | Chimpanzee | P. troglodytes | Great ape | S003611 | Skin, location unknown | Male | 6 yr | 1 |
| Donor 3 | Chimpanzee | P. troglodytes | Great ape | S003649 | Skin, location unknown | Male | 9 yr | 1 |
| Donor 1 | Rhesus macaque | M. mulatta | Old World Monkey | AG08308 | Skin, arm | Male | 1 yr | 1 |
| Donor 2 | Rhesus macaque | M. mulatta | Old World Monkey | AG08312 | Skin, arm | Female | 1 yr | 1 |
| Donor 3 | Rhesus macaque | M. mulatta | Old World Monkey | AG08305 | Skin, arm | Male | 1 yr | 1 |
| Donor 1 | Human | Homo sapiens | Great ape | NHDF | Skin, location unknown | Female | 1 yr | 7 |
| Donor 2 | Human | H. sapiens | Great ape | NHDF AF | Skin, location unknown | Male | 1 yr | 6 |
| Donor 3 | Human | H. sapiens | Great ape | NHDF R06 | Skin, location unknown | Male | 1 yr | 5 |
| Donor 1 | Mouse | Mus musculus (C57/BL6) | Rodentia | C57A | Skin, abdomen | Male | 8–10 wk | 1 |
| Donor 2 | Mouse | M. musculus (C57/BL6) | Rodentia | C57B | Skin, abdomen | Male | 8–10 wk | 1 |
| Donor 3 | Mouse | M. musculus (C57/BL6) | Rodentia | C57C | Skin, abdomen | Female | 8–10 wk | 1 |

relatives but also providing additional sequencing information from NHP species, including those that are endangered. In this study, we mapped the data from each species both to the human genome and to the species-specific genomes, creating the added opportunity of comparing these two approaches in parallel. Overall, we observed a high level of conservation in the functional responses of the NHP DFs to poly(I:C) and found that with the current state of genome annotations, the percent of reads assigned to a genetic feature were largely similar between the two mapping methods. We anticipate that making these data available will enhance the continued efforts to more fully annotate NHP genomes and to aid in the identification of novel transcripts unobserved in humans. In addition, we believe these unique data will greatly facilitate evolutionary analyses, especially in the area of comparative immunology.

# Results and Discussion

## Expression of select interferon-stimulated genes (ISGs) in response to poly(I:C) varies across diverse species

Triggering innate immune sensing pathways characteristically provokes the induction of hundreds of ISGs, which collectively create an antiviral state. However, it remains unclear how similar such responses are across NHP species. To initially test whether poly(I:C) would elicit a cell-intrinsic response in the DFs we had from 10 different species (each represented by three donors), we first assessed transcription via RT-qPCR of *IFNβ* at 12 and 24 h post-transfection and two well-characterized ISGs, myxoma resistance protein 1 (*MX1*) and 2′-5′-oligoadenylate synthetase-like (*OASL*, *Oasl1* in mice), 24 h post-transfection using species-specific primers (Figs 1C–E, S1, and S2). *IFNβ* mRNA had on average less than a 10-fold change in expression relative to mock-transfected cells for most species (Fig 1C). However, bonobo, gorilla, orangutan, and squirrel monkey did exceed this for at least one of the time points tested. Bonobo and gorilla had significantly higher *IFNβ* expression at 12 h compared with 24 h, whereas the opposite trend was observed for rhesus macaque. The two ISGs examined demonstrated far greater fold changes relative to mock-transfected cells, underscoring the prominence of the downstream response to IFN at 24 h post-transfection. All species, with the exception of mouse, had at least an average 100-fold increase relative to mock in *MX1* mRNA, although there was marked variation amongst rhesus donors. Average levels of *OASL* mRNA were also elevated across the primate donors, albeit with greater donor variation, but all three mouse donors still displayed minimal change. The donor–donor variation we observed was not surprising as the NHP fibroblasts were acquired from outbred donors, with at least one male and one female represented for each species. As these data represent only a small sample of the hundreds of ISGs (28, 29, 30) whose expression could be changing after poly(I:C) transfection, we performed RNA-Seq on total mRNA isolated from these DFs 24 h post-transfection for a more comprehensive view. Unlike recent studies that looked earlier at 4 h post-immune stimulation (26, 27), we wanted to analyze effects further downstream as the cell-intrinsic response was amplified over time.

## Using species-specific genomes increases read alignment but to regions with no assigned features

In working with such a diverse array of species, one challenge was choosing the genomes upon which to map the NHP-derived RNA-Seq reads. Since the genomes for human and mouse are well-annotated and species-specific resources available for downstream analyses, we felt confident in aligning reads from these species to their respective genomes. However, for the NHP species, to make our analysis more thorough, we used two approaches in parallel: mapping all primate-derived reads to the human genome as previously performed (31, 32, 33, 34) (hereby referred to as "human method") or to their respective genomes as they currently exist on Ensembl (Table 2) ("species method"). Human and mouse reads averaged above 90% alignment and ~80% assignment. For the great ape species examined, the percent alignment was generally the same regardless of the genome used, exceeding 85% (Fig S3). For the Old World monkeys, there was a small but noticeable decline when mapping to the human genome, with percent alignment close to 85% for the macaques and dipping under 80% for samples from olive baboon. Aligning reads from each of these species to their respective genomes increased these values, especially for rhesus and pig-tailed macaque, which approached 90% and 95%, respectively. Moving further out evolutionarily, squirrel monkey had the lowest percent alignment to the human genome, but still most reads did map, centering around 70%. Not surprisingly, more than 90% of reads were aligned when the squirrel monkey genome was used.

Despite these variations in alignment depending on the reference genome used, the percent of reads assigned to a genome feature by the "human method" tended to be higher (up by ~5–6%) or comparable with that of the "species method." Only for some chimpanzee, pig-tailed macaque and squirrel monkey samples did the species mapping improve the percent assignment (once more up by ~5–6%). Orangutan and olive baboon demonstrated the greatest differences, with the "species method" as much as ~10 or 15 percentage points lower, respectively, than the "human method."

**Table 2. Reference genomes used.**

| Species | Reference genome | Last genebuild update |
|---|---|---|
| Orangutan | ppyg2 (Ensembl 93) | August 2012 |
| Gorilla | gorGor4 (Ensembl 95) | January 2018 |
| Pig-tailed macaque | Mnem_1.0 (Ensembl 95) | January 2018 |
| Rhesus macaque | Mmul_8_0_1 (Ensembl 95) | October 2016 |
| Bonobo | panpan1.1 (Ensembl 95) | January 2018 |
| Chimpanzee | Pan_tro_3.0 (Ensembl 95) | January 2018 |
| Olive baboon | papAnu2 (Ensembl 95) | January 2018 |
| Squirrel monkey | SaiBol1.0 (Ensembl 95) | January 2018 |
| Human | GRCh38 (Ensembl 96) | November 2018 |
| Mouse | GRCm38 (Ensembl 85) | May 2016 |

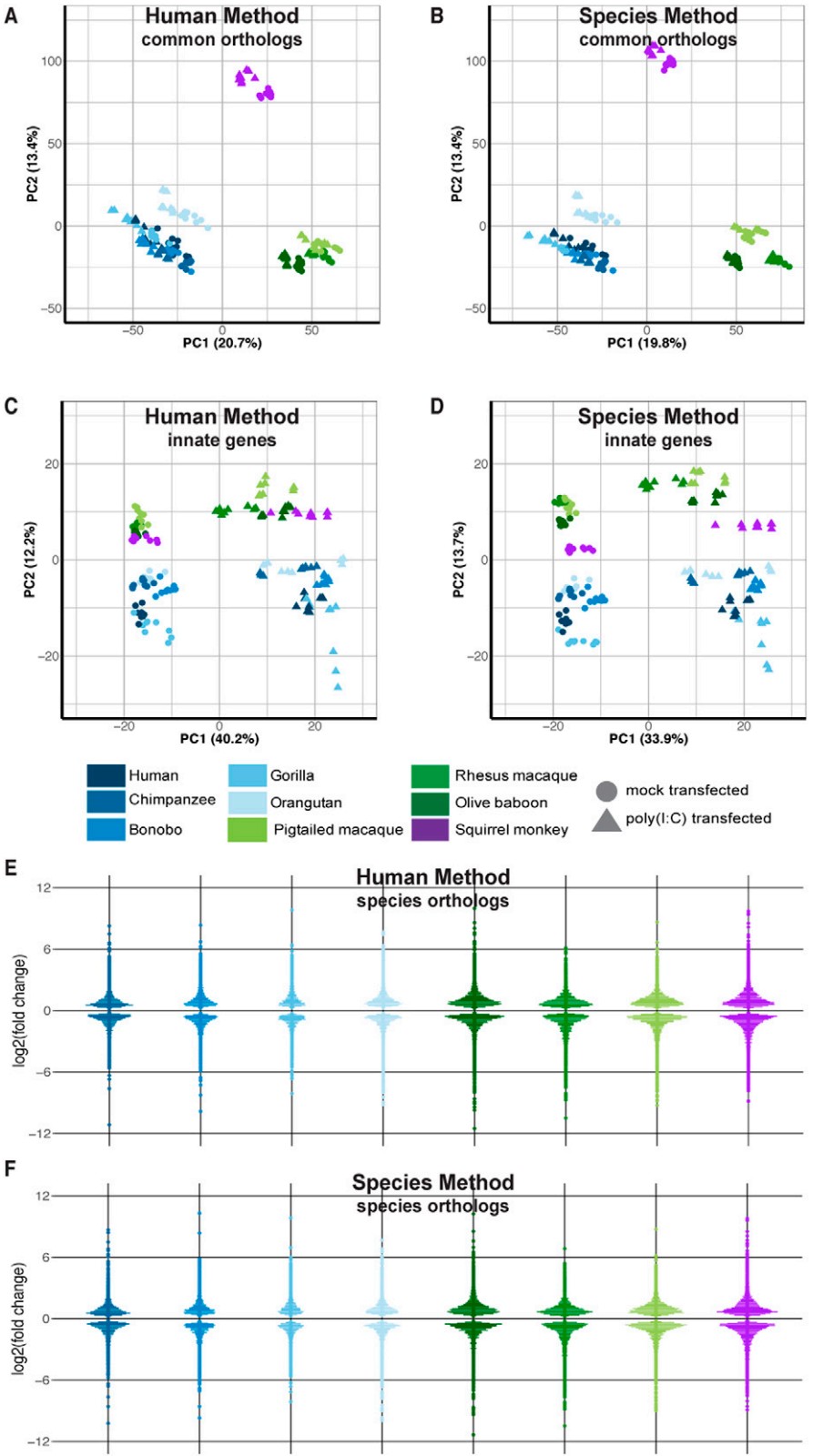

**Figure 2. Overall transcriptomic responses to poly(I:C) cluster by species' phylogenetic relationships and innate immune genes by treatment condition.**
**(A, B, C, D)** RNA sequencing reads were aligned to the human genome (A, C) or the respective NHP species from which the samples were derived (B, D). The resultant counts from either mapping method were then limited to genes with a one-to-one human ortholog (based off Ensembl ID) across all species, resulting in ~11,000 genes (referenced as "common orthologs" in the main text). For counts mapped to the NHP genomes, the human ortholog Ensembl ID was used for all subsequent analysis for ease of comparison. These counts were normalized using DESeq2 default options and finally transformed using the regularized $\log_2$ function in DESeq2. **(A, B, C, D)** The first two components of the PCA are depicted here for either all ~11,000 genes ("common orthologs") (A, B) or for a subset of these genes which overlapped with those found in the database InnateDB ("innate genes") (C, D). Each point represents an individual donor and is colored by species as shown in the figure legend, with shades of blue indicating great ape species, shades of green Old World Monkeys, and purple the single New World Monkey used in this study. Circles indicate mock-transfected samples and triangles poly(I:C)-transfected samples. **(E, F)** The counts from mapping to either the human or the species-specific genome were limited on a species-by-species basis to the Ensembl IDs that had a one-to-one human ortholog (referenced as "species orthologs" in the main text). The counts of these genes were then processed in DESeq2 to determine DGE, specifically comparing the DGE profile of each NHP species (poly(I:C)-transfected versus mock-transfected) to that of human. Thus, genes from NHP species that significantly differed from human in their response to poly(I:C) could be determined and are shown here as pseudo-violin plots from either the human genome (E) or the species-specific genome (F) mapping method. Each point represents an individual gene and is colored by species as shown in the figure legend. Data information: in (E, F), the differentially expressed genes shown have $P_{adj} \leq 0.05$.

Taking a closer look at the relationship between the percent of reads assigned to a genome feature and the mapping method, we observed that the benefit of the "species method" over the "human method," whereby the percentage of reads failing to be assigned a feature because of poor mapping quality declined, was less advantageous than expected as an increased percentage of the reads that

mapped were to regions with no assigned feature (Fig S4). Notably, this was consistent across NHP species, irrespective of evolutionary relationship to humans. Thus, the percentage of reads assigned a feature was generally comparable between the two mapping methods or better by the "human" method. The NHP genome annotations as they currently exist resulted in on average anywhere from ~12–22% of reads mapping to sequence with no assigned feature (for comparison, <5% of human and mouse reads fell in this category after alignment to their respective genomes). In contrast, less than 5.5% of reads fell in this category when using the "human method." Identifying what these unassigned features might be beyond the extensive homologous sequence searching across existing genomes already performed by Ensembl is outside the scope of this article. These data highlight the weaknesses of the current NHP genome annotations and the core difficulty in performing comparative analyses, as the more divergent, species-specific sequences will still be missed regardless of the genome used. We hope that sequencing datasets like ours from multiple donors of multiple NHP species will aid in improving such annotations.

### Species cluster by evolutionary relationship when looking at overall transcriptomic response

Across all NHP species examined, most genes had a one-to-one human ortholog, with the second largest category of genes those without a known ortholog (Fig S5). For both the "human" and the "species" methods, we first limited the resultant mapped read counts to only those genes that had one-to-one orthologs across all species ("common orthologs"), resulting in a common denominator of 11,677 genes (Supplemental Data 1). Using such an approach, we were able to directly compare all eight NHP species with one another, finding that by principal component analysis (PCA), the samples clustered according to the evolutionary relationship of the species, with great apes, Old World monkeys, and the single New World monkey species forming their own distinct groupings regardless of whether they were mapped by the "human" or the "species" method (Fig 2A and B). For both methods, the orangutan samples formed a group slightly removed from the other great ape species. However, only by the "species" method did we observe resolution of the Old World monkey species into three distinct groups unlike by the "human" method. This could be, at least in part, attributed to the larger differences in percent assignment of reads, especially between olive baboon and the two macaques, when using the "species" method (Fig S3).

### Innate immune genes strongly up-regulated by poly(I:C) transfection across species

As we had transfected the cells with poly(I:C), a synthetic dsRNA mimic known to induce cell-intrinsic immune responses, we wanted to focus further on known innate immunity genes and how their expression compared across these primate species. We limited our "common orthologs" output to a manually curated set of genes involved with innate immunity, InnateDB (35). This database includes 988 distinct gene symbols for human, 656 of which fell into our "common orthologs" set as one-to-one orthologs across all species (Supplemental Data 2). Examining these genes across our samples by PCA, we now observed clear separation of samples

based on whether they had been transfected with poly(I:C) (Fig 2C and D, mouse shown in Fig S6), indicating the strong contribution of treatment in differentiating samples based off these genes. Within the treated and mock samples, the great ape samples did cluster separate from the monkey species, although mapping by the "species method" did result in greater separation between Old and New World monkey species. The squirrel monkey samples, our only New World monkey species under consideration, were closer than the Old World monkeys to the great ape cluster, especially after the "species method," suggesting higher similarity of these genes' expression with the great apes even though they are more distantly related.

We examined the sample-to-sample distances of these same data to see if we still observed a closer relationship between the great apes and squirrel monkey when not limited to just the first two principal components. Hierarchical clustering of the sample-to-sample distances was highly similar regardless of mapping method, clearly distinguishing between mock and poly(I:C)-transfected samples as well as the distinct nature of poly(I:C)-transfected samples from gorilla donor PR00107 (Fig S7). In addition, despite the PCA plot for the "human method" showing less separation between the New and Old World monkeys, we were able to confirm the greater similarity of the squirrel monkey samples to the great apes in regards to this gene set (Fig S7). Although squirrel monkey was the most distant of our NHP species from humans, the clustering of this species with the great apes, instead of out beyond the Old World monkeys, suggests strong conservation of the response in these innate immune genes after poly(I:C) transfection. However, the exact basis for these observations cannot be determined, especially without examining more New World monkey species to see if this is broadly applicable.

### Determining the NHP response to poly(I:C) distinctive from that of human

Although informative for broader analyses, using the genes with a one-to-one human ortholog across all eight species was far more limiting than if we did so on a species-by-species basis (referred to as "species orthologs"). Such an approach substantially increased the possible number of Ensembl IDs from 11,766 to 15,793-22,040 (Table 3; note that after determining the one-to-one human ortholog for a given NHP gene, the human Ensembl ID was used). In

**Table 3. Number of Ensembl ID annotations for each NHP species that is listed as having a one-to-one human ortholog.**

| Species | Number of one-to-one orthologs with human |
|---|---|
| Chimpanzee | 21,766 |
| Bonobo | 22,040 |
| Gorilla | 21,998 |
| Orangutan | 19,843 |
| Olive baboon | 20,506 |
| Rhesus macaque | 19,680 |
| Pig-tailed macaque | 17,058 |
| Squirrel monkey | 15,793 |

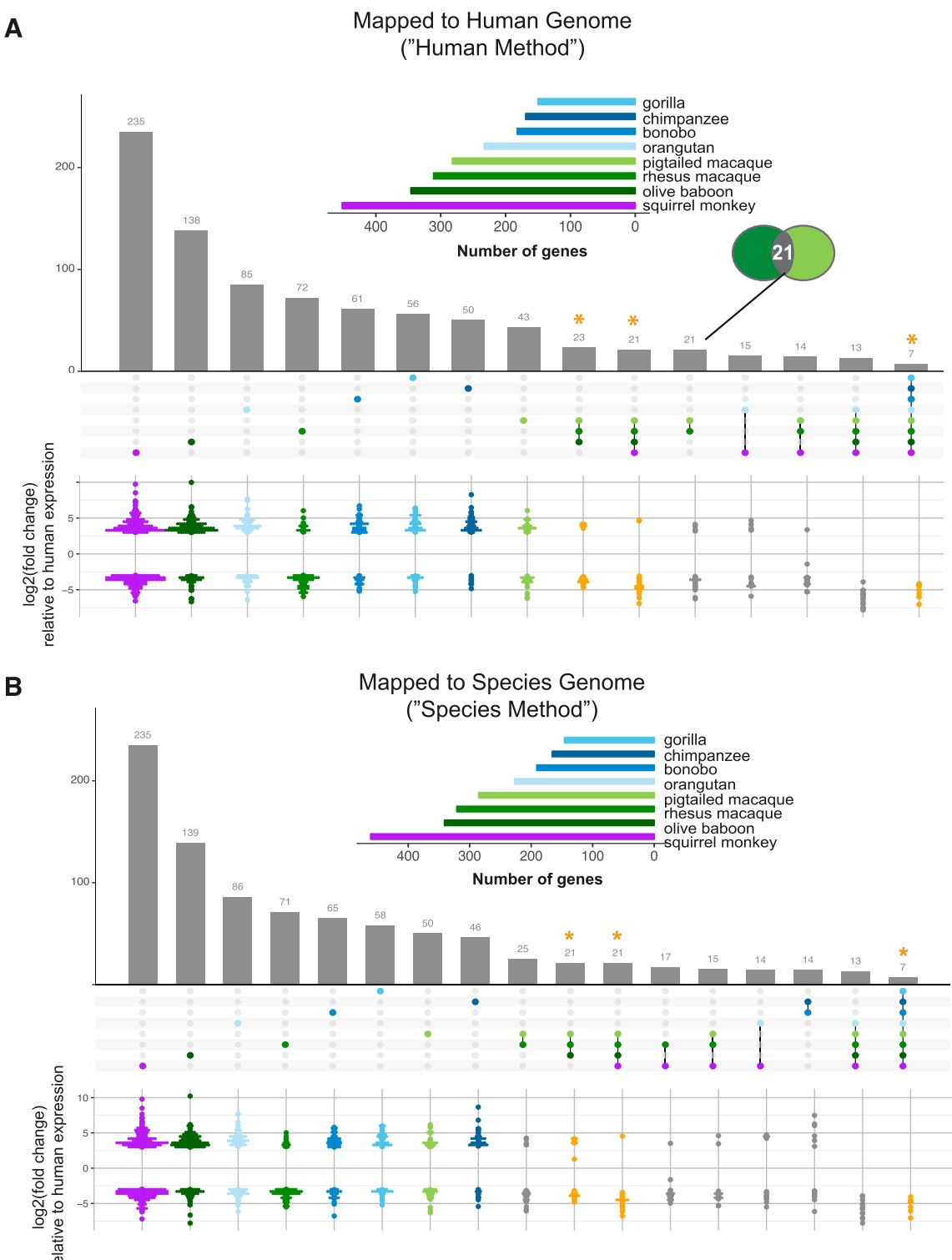

**Figure 3. Overview of genes in NHP species and subgroups that differ markedly in their response to poly(I:C) compared with that of human.**
**(A, B)** For the genes shown in Fig 2E and F, the RNA-Seq reads were mapped to either the human genome (A) or the species-specific genome (B). The resultant gene counts from each mapping method were then limited on a species-by-species basis to those for which there is a one-to-one human ortholog. DGE upon poly(I:C) transfection for each of the NHP species was then assessed relative to that of human. The vertical bar graph in the upper panel displays the number of genes which were differentially expressed ($P_{adj} \leq 0.05$, |$\log_2$(fold change)| ≥ 3) by a given species or group of species, as indicated by the matrix below the graph, compared with human. Each bar, thus, represents a unique group of genes with the matrix beneath indicating in which species these genes are expressed, similar to what is represented by the "intersection" of circles in a Venn diagram (as indicated by the inset Venn diagram in (A)). For ease of presentation, only the intersections with 14 or more genes are shown, except for the intersection of seven genes that differ from human in their expression upon treatment in all the species. The points

looking at the differential gene expression (DGE) profiles (poly(I:C)-transfected versus mock-transfected) for each species using "species orthologs," we observed similar distributions, with more genes increasing versus decreasing in expression after poly(I:C) transfection compared with mock-transfected cells (Fig S8, and Supplemental Data 3 and 4). Although the range of expression for up-regulated genes was larger compared with down-regulated genes, most genes had a $|\log_2(\text{fold change})|$ <3 relative to mock-transfected cells. Because we were most interested in how the NHP species compared with human, we used the latter as our baseline to assess DGE for each species ($P_{adj}$ ≤ 0.05), finding on average ~2,800 genes differentially expressed compared with human among the great ape species and ~4,800 among the monkeys (Fig 2E and F, and Supplemental Data 5).

To find genes in each species with markedly different expression in response to poly(I:C) from that of human, we limited these DGEs to those with a $|\log2(\text{fold change})|$ ≥ 3 and then compared their occurrence across the species. By either mapping method, more genes overall were differentially expressed compared with human in accordance with the evolutionary distance of the species from human (Fig 3, horizontal bar graphs). Furthermore, ~80–89% of these genes overlapped between the two mapping methods (Table 4). A fraction of these genes were unique to each NHP species (Fig 3, first eight columns), with proportionally the greatest in two of the species more distantly related to human: squirrel monkey and olive baboon. By either mapping method, most genes unique to a given species had higher expression relative to human except for rhesus macaque, where the opposite trend was observed, and squirrel monkey and pigtailed macaque, which each had an almost equal number of genes with higher and lower expression (Fig 3, pseudo-violin plots). In all cases, most genes found as uniquely different from human in a given species or species grouping were one-to-one orthologs across all species (Fig S9). Thus, these genes' status as "unique" was not simply a result of their exclusion by our "species orthologs."

In addition, of the possible combinations of species with common genes differentially expressed compared with human, we focused on NHP groupings of evolutionary significance (Fig 3, columns with orange asterisks): Old World monkeys, all monkey species, and all species (there were no genes common to all the great apes). For these genes, the average differential expression relative to human was determined for the species in each grouping and tended by either mapping method to be lower than that of human (Fig 3, pseudo-violin plots). Our finding that of the 3,000–6,000+ genes differentially expressed ($P_{adj}$ ≤ 0.05) upon poly(I:C) treatment in the NHPs <500 were differentially expressed compared with human at our cutoff highlights the similarity of these species with human.

For all these genes differentially expressed compared with human in individual NHP species or groups of species, we then examined the DGE profiles we had generated of poly(I:C) versus mock-transfected cells for each species (Supplemental Data 3 and 4). Thus, we aimed to find the basis for each gene's significantly different expression compared with human. For example, genes with a negative $\log_2(\text{fold change})$ in comparing humans versus NHPs could be because the gene's expression was unchanged by poly(I:C) transfection in NHPs but increased in humans, was decreased upon poly(I:C)-transfection in NHPs but less so or not at all in human, or did increase in NHPs after poly(I:C) transfection but at a magnitude lower than that for human. We prepared heat maps to answer this question for the genes "unique" to a given species (Fig S10) and then the three groups of interest (Old World monkeys, all monkeys, and all species) after mapping by the "human method" (Fig 4A) or the "species method" (Fig 4B). For these three groupings, regardless of mapping method, most genes that strongly increased upon poly(I:C) transfection in humans showed weak changes that were often nonsignificant in the NHP species for that group (Supplemental Data 6). With a few exceptions, when a gene's expression was significantly different between treatment conditions in the NHP species, the expression of the NHP genes tended to follow the same trend as humans but at a weaker magnitude. In the smaller group of genes different across all species compared with human, all the genes except one demonstrated decreased or negligible change in expression after poly(I:C) transfection in the NHP species but increased expression in humans. The exception, an InnateDB gene phosphoinositide-3-kinase adaptor protein 1 (*PIK3AP1*; plays a role in immune cell development and controlling cytokine production), stood out as being up-regulated, albeit at a lower magnitude than in humans, in squirrel monkey, gorilla, and bonobo but negligibly changed in the remaining species. Between mapping methods for these smaller subgroups, most genes overlapped.

For the genes "unique" to each of the great ape species, most were significantly different upon poly(I:C) transfection, whereas for these same genes in humans, the change was negligible (Fig S10 and Supplemental Data 6). The number of nondifferentially expressed genes was more evenly split between human and each of the macaque species (Fig S10 and Supplemental Data 6). In accordance with the summary data in Fig 3, most genes in the rhesus samples had lower $\log_2(\text{fold change})$ values compared with humans, with up-regulation occurring in humans but nonsignificant or low-level increases in rhesus. The higher number of genes for squirrel monkey and olive baboon is likely reflective of their more distant relationship to human (Fig S10 and Supplemental Data 6). For both of these NHP species, there was a subgroup of genes whose expression was clearly up- or down-regulated after poly(I:C) treatment that were

in the matrix correspond to the different species (color-coded) from which DFs were sourced, with shades of blue indicating great ape species, shades of green Old World Monkeys and purple the single New World Monkey used in this study. The inset horizontal bar graph shows the total number of differentially expressed genes for each species based on the cutoff used ($P_{adj}$ ≤ 0.05, $|\log_2(\text{fold change})|$ ≥3). The pseudo-violin plot below the matrix summarizes the differential $\log_2(\text{fold change})$ expression relative to human for the genes included in the corresponding vertical bar. For genes expressed by multiple species (i.e., all except the first eight columns), the average $\log_2(\text{fold change})$ across species for each gene was used. For the first eight columns of genes unique to each species, the dots are colored in accordance with the horizontal bar graph. An orange asterisk with corresponding orange points in the pseudo-violin plot highlights the gene groups of particular interest that are in common across larger NHP groupings: all Old World monkeys, all monkeys, or all species.

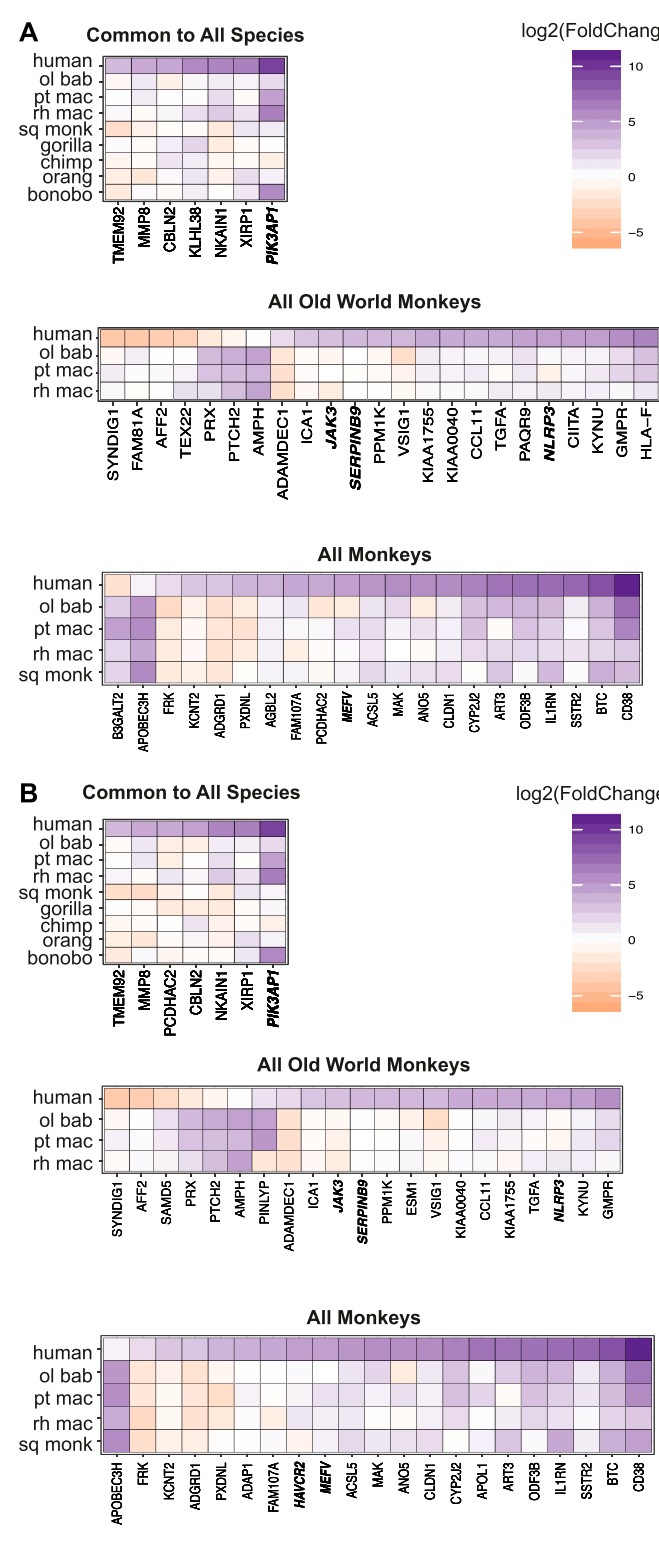

**Figure 4. Differential expression profiles on a species-by-species basis for genes significantly different from human in subgroups of NHP species after poly(I:C) transfection.**
**(A, B)** For the genes found to be significantly differentially expressed compared with humans ($P_{adj}$ ≤ 0.05, |log₂(fold change)| ≥ 3) among all Old World monkeys, all monkeys, or all species (as shown in Fig 3), the DGE between poly(I:C)-transfected and mock-transfected samples was determined for each NHP species

nonsignificantly or negligibly different in the human samples. Likewise, the opposite trend was seen at either extreme of the human genes, with squirrel monkey and, to a lesser extent olive baboon, showing smaller or nonsignificant changes in genes that were strongly up- or down-regulated in human.

### Enrichment for cell-intrinsic immune response pathways conserved across all species

Biological outcomes are not necessarily dictated by individual genes but rather the concerted efforts of multiple genes acting in tandem. Although innate immune genes were present amongst genes differentially expressed in NHP but not in human samples, it was unclear if this handful of players was resulting in large-scale differences in the cell-intrinsic immune responses of these various species to poly(I:C). Furthermore, as only one time point post-transfection was tested in this study, it is impossible to determine if genes with altered expression from humans are differentially stimulated by poly(I:C) or the result of varying transcriptional kinetics. For a broader overview of these DGE profiles, we performed gene set analysis, whereby data are assessed for an enrichment of genes related in some way, such as their participation in the same pathway or biological function (36, 37, 38, 39). Because each species was limited to its own number of one-to-one orthologs, we prepared a human DGE profile for each species limited in the same way to ease comparisons. We used the R package Generally Applicable Gene-set Enrichment (GAGE) (40) to determine the Kyoto Encyclopedia of Genes and Genomes (KEGG) (41, 42, 43) pathways with the most significantly altered gene expression after poly(I:C) transfection. To reflect the nature of pathways in biological systems, bidirectional changes in gene expression were considered when determining pathway enrichment. For each NHP species besides rhesus macaque and regardless of the mapping method, genes were significantly enriched ($q_{val}$ ≤ 0.09) for multiple pathways known to be important for cell-intrinsic immune responses, either directly or indirectly, including Toll-like receptor (TLR) signaling, nucleotide-binding oligomerization domain (NOD)-like receptor signaling, cytosolic DNA sensing, JAK-STAT signaling, antigen processing and presentation, and chemokine signaling (Fig 5; the latter two were also significant for mice, Table 5). Although all the NHPs had pathway profiles that corresponded well with the accompanying human profile, rhesus macaque failed to meet the

and is shown with the accompanying expression profile for the same genes in human. Because read counts were mapped to either the human (A) or the species-specific genome (B) and the genes limited on a species-by-species basis to the Ensembl IDs that had a one-to-one human ortholog (referenced as "species orthologs" in the main text), each NHP species has a corresponding human DGE profile limited to those same genes. Thus, the human profiles were generated from the same set of biological samples just limited to different lists of genes for each species. For presentation purposes of the multispecies groups, the human values shown are from the olive baboon ortholog-limited expression profile. Because these genes were found across multiple species, the expression pattern only marginally differs between the various accompanying human profiles. Genes that are decreased in expression upon poly(I:C) treatment relative to mock are shown in orange and those increased in purple, with the intensity of the color corresponding to the magnitude of the change (expressed as log₂(fold change)). The names of genes which are from the InnateDB list are shown in bold italics.

**Table 4. Overlap of genes between mapping methods for each NHP species significantly different in their transcriptomic response to poly(I:C) compared with that of human.**

| NHP species being compared with humans | No. of genes from mapping to species-specific genome[a] | No. of genes from mapping to human genome[a] | No. of genes in common between two mapping methods | Percentage of species genome–mapped genes in common between methods | Percentage of human genome–mapped genes in common between methods |
|---|---|---|---|---|---|
| Bonobo | 191 | 182 | 153 | 80.10 | 84.07 |
| Chimpanzee | 166 | 169 | 143 | 86.14 | 84.62 |
| Gorilla | 146 | 150 | 127 | 86.99 | 84.67 |
| Olive baboon | 341 | 346 | 276 | 80.94 | 79.77 |
| Orangutan | 227 | 233 | 194 | 85.46 | 83.26 |
| Pig-tailed macaque | 285 | 282 | 252 | 88.42 | 89.36 |
| Rhesus macaque | 321 | 311 | 276 | 85.98 | 88.75 |
| Squirrel monkey | 461 | 452 | 396 | 85.90 | 87.61 |

[a]Genes in treated-versus-mock NHP samples that differed from treated-versus-mock human samples by abs[$\log_2$(foldchange)] ≥ 3, $P_{adj}$ ≥ 0.05

significance cutoff for any pathway. This was not due to limiting the list of genes to one-to-one orthologs as evidenced by the accompanying human profile, which was limited to the same gene list. The top-ranking pathways were the same as the other species but likely did not pass the significance threshold because of the lower number of differentially expressed genes relative to the other species. In some instances, such as NOD-like receptor signaling for olive baboon and retinoic acid–inducible gene I–like receptor signaling for bonobo, the significance value fell just outside of the cutoff, indicating that although perhaps not to the same extent as the other NHPs, these pathways are still important. Apoptosis and osteoclast differentiation genes stood out as clearly significant in their enrichment only for squirrel monkey and some of the corresponding human profiles. Although apoptosis is less unexpected, osteoclast differentiation could be appearing as a result of the common origin of osteoclasts and immune cells from hematopoietic cells and the role of TLR stimulation in disrupting osteoclast differentiation (44).

Together, these data mimicking a viral infection serve as a first step in comparing the cell-intrinsic immune responses of the most diverse panel of NHP species to date. In using a synthetic compound as a proxy for RNA virus infection, we were able to analyze transcriptomic responses without having to account for species-specific permissiveness and/or susceptibility to a pathogen. While there were of course differences in individual gene expression profiles after poly(I:C) transfection across species, these ultimately resulted in similar outcomes, activating pathways important to cell-intrinsic immunity. We compared our findings to a recent study of rhesus, mouse, and human primary DFs transfected with poly(I:C) for 4 h (26). Although our time point was 20 h later, some of these early genes (filtered for primates to those with a one-to-one human ortholog) remained up-regulated in these same species at 24 h ($P_{adj}$ ≤ 0.05, $\log_2$(foldchange) ≥ 3) – ~22% for rhesus, ~38% for mouse, and ~60% for human (Supplemental Data 7). This comparison highlights the rapid nature of cell-intrinsic immune responses and the sustained transcription of these responses promoted by continued ISG production. However, it also suggests species–species differences in the kinetics of such responses that will be an important area for

further research. Similarly, among the 62 genes that were commonly increased in response to type I IFN treatment in species far more divergent from the ones we studied (bats, chicken, horse, sheep, rat, cow, human, and pig) (27), from 50% (rhesus) to 77% (squirrel monkey) of the genes that had a one-to-one human ortholog in each of our NHP species (>90% of the 62 genes) were also up-regulated 24 h post-poly(I:C) treatment using the same cutoffs as those of Shaw et al ($P_{adj}$ ≤ 0.05, $\log_2$(foldchange) ≥ 2; Supplemental Data 8). In comparison, our human samples showed ~80%.

Although we observed a high level of similarity across these NHP species, this is not to say that in the context of a viral infection, all species would mount the same response. In transfecting the cells with poly(I:C), we provoked a response that is not tempered by the antagonism of viral proteins and thus closer to what the potential cell-intrinsic immune response of these cells would be if left uninhibited by the actions of viral proteins. As a result, the data presented here do not presume to predict the susceptibility of these species to particular viral pathogens or whether other cell types from these species would react the same way. We tested the assumption that the species' response to poly(I:C) would be highly similar as this had not previously been formally demonstrated for all of these species. Thus, although striking differences between species would have been of great interest, a quantitative demonstration of their similarity is still very useful. Knowing the unfettered capabilities of these cells in terms of cell-intrinsic immunity provides a solid foundation to explore how this more maximal response is potentially dampened by the interplay of host and viral proteins.

In addition, as alluded to above, because we only examined one time point, the kinetics of the transcriptomic response provides another area for future analysis that could demonstrate interesting differences across species. These data will serve as an important baseline for such work, showing what transcriptional responses are possible and how they are impacted upon exposure to a pathogen. Furthermore, because the current study only examined poly-A–containing RNA transcripts, we cannot exclude the role of other RNA species, such as long non-coding RNAs that are becoming of increasing interest in NHP genomics ((45), reviewed in reference 46).

**A**

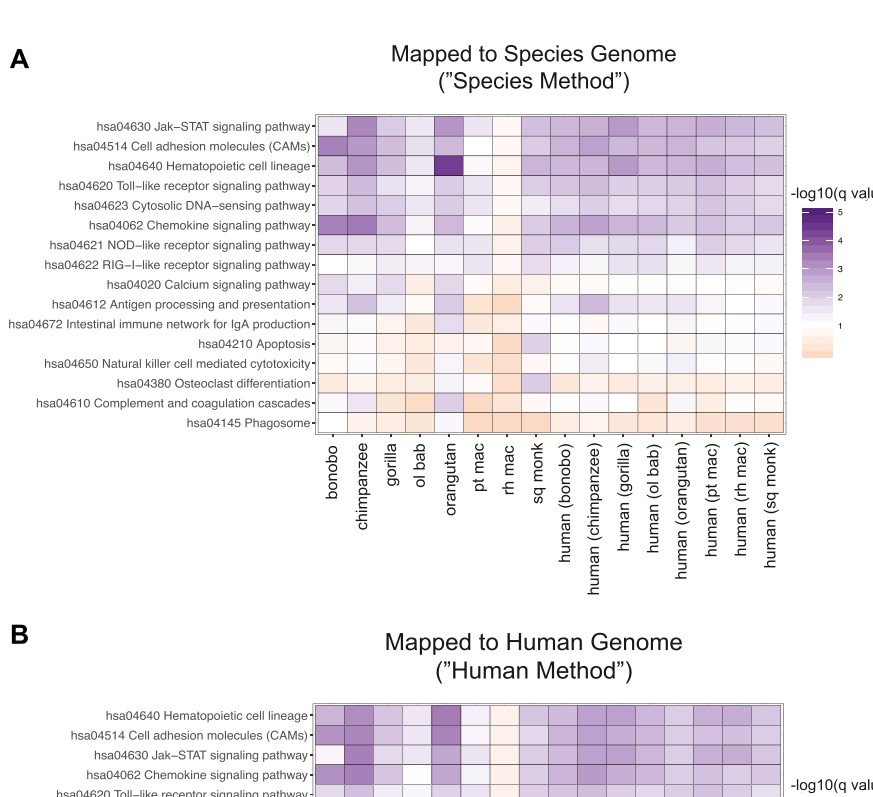

Mapped to Species Genome
("Species Method")

**B**

Mapped to Human Genome
("Human Method")

**Figure 5. Enrichment of genes involved with cell-intrinsic immune pathways dominate the response to poly(I:C) across all species.**
**(A, B)** Read counts mapped to the human (A) or species-specific genomes (B) were limited on a species-by-species basis to the Ensembl IDs that had a one-to-one human ortholog (referenced as "species orthologs" in the main text). The resultant genes were then analyzed using GAGE, with the only genes excluded being those that lacked an Entrez ID (the IDs used by GAGE) and/or had a missing $P_{adj}$ value. A heat map is shown of pathways (the row labels) with a $q$ value ≤ 0.09 for at least one species (the column labels), with the cells colored by $-\log_{10}(q$ value) so that cells meeting the cutoff are shades of purple and those that do not are shades of orange, with white set at the cutoff of $-\log_{10}(0.09)$. Because genes were limited on a species-by-species basis to those that had a one-to-one human ortholog, each NHP species has a human profile that was limited to that same gene set (the eight right most columns of each heat map). The reads were derived from the same experimental samples—only the downstream analysis differs in terms of the genes that were included by their ortholog status.
Source data are available for this figure.

Finally, our parallel workflow allowed us to compare the benefits of mapping to the human versus the species-specific genome for these NHP species. The broader outcomes, such as the pathway enrichment analysis, were strongly similar between the two methods, and as noted earlier, the additional alignment gained by using species-specific genomes was often to regions with no features yet assigned. As a further comparison of the two mapping methods, we examined the top 500 most significantly differentially expressed genes ultimately identified after either the "species method" or the "human method." Depending on the species, 88–93% of the top 500 most significant genes overlapped between the two mapping methods (Fig S11 and Supplemental Data 9). For each set of 500 genes from a given mapping method, we compared the $\log_2$(fold change) and significance for those same genes from the alternative mapping method. The $\log_2$(fold change) values were highly correlated and we observed for each species less than 15 genes that were (i) significantly differentially expressed by one

approach ($P_{adj}$ ≥ 0.05) but not the other or (ii) had a $\log_2$(fold change) value that was 1.5 times higher or lower in one mapping method versus the other and also had a $|\log_2$(fold change)| ≥ 2 in at least one of the mapping methods. For genes that met these criteria, the differing expression between mapping methods was due to a variety of reasons. Reads which mapped to a region that had overlapping annotations in one of the reference genomes but not the other caused those reads to be excluded for not mapping to a unique feature. In other instances, multiple transcripts of the same gene were annotated covering a large region in one reference but not the other, so although alignment was observed by either mapping reference, an annotation only existed for one of the genomes.

These findings are in line with the general observation that the human genome is more well annotated than those of NHPs and underscores the importance of improving the feature assignment of these sequences. RNA-Seq datasets such as ours can help

**Table 5. Top 12 pathways for which genes are enriched in mice after poly(I:C) transfection.**

| Kyoto Encyclopedia of Genes and Genomes pathway | q value |
| --- | --- |
| mmu04514 cell adhesion molecules | 0.005223675 |
| mmu04612 antigen processing and presentation | 0.076815675 |
| mmu04062 chemokine signaling pathway | 0.076815675 |
| mmu04620 TLR signaling pathway | 0.101942397 |
| mmu04672 intestinal immune network for IgA production | 0.112781502 |
| mmu04623 cytosolic DNA-sensing pathway | 0.112781502 |
| mmu04621 NOD-like receptor signaling pathway | 0.112781502 |
| mmu04145 phagosome | 0.112781502 |
| mmu04610 complement and coagulation cascades | 0.232711025 |
| mmu04640 hematopoietic cell lineage | 0.235522039 |
| mmu02010 ABC transporters | 0.235522039 |
| mmu04622 RIG-I-like receptor signaling pathway | 0.252956436 |

identify novel exons or in some cases aid in rectifying feature assignments at regions where exons were incorrectly annotated. For example, as alluded to above, a given gene in the human genome may have more annotated transcript variants that can sometimes cover a larger region and lead to read assignment, whereas the orthologous NHP gene, with sometimes only one listed variant, has no reads assigned to it as the reads map but to an area outside of the limited region where the single variant was annotated. In making these data available, we anticipate that they will facilitate continued efforts to annotate the genomes of NHP species and to identify additional transcript variants of already annotated genes as well as completely novel transcripts unobserved in humans. As with the "1,000 Genomes Project" for human samples (47), having more sequencing data available from multiple individuals for a given species aids in identifying single-nucleotide polymorphisms and other sources of genomic variation to improve annotations. We explicitly strove to include at least one male and one female with varying ages across the three donors. This is far more reflective of the natural variance that exists in any outbred species population and makes our sequencing reads all the more useful, especially for genomes, such as that of orangutan, sourced from a single individual. We hope our present analysis and this data collection more broadly will serve as a springboard for additional evolutionary analysis and comparisons.

# Materials and Methods

### DFs

Mouse DFs from C57BL/6 mice were purchased from Cell Biologics. Primary human DFs were obtained from American Type Culture Collection (PCS-201-012). All the NHP DFs were purchased from the Coriell Institute for Medical Research with the exception of two squirrel monkey donors (SQMA and SQMB), which were derived from skin biopsies generously provided by Robert Lanford (Southwest Biomedical Research Center). For any of the DF donor

species that are on the United States Fish and Wildlife Services endangered or threatened species list, the Coriell Institute has the appropriate and required documentation of breeding records (Supplemental File 1) indicating captivity before or birth after November 18, 1976. The squirrel monkey skin biopsies were obtained after terminal necropsy using Princeton University Institutional Animal Care and Use Committee–approved protocols (#1930) and shipped overnight on wet ice to Princeton University. Upon arrival, the skin was prepped and DFs isolated according to previously published protocols (48). In brief, the skin biopsies were scraped to remove connective tissue, cut into smaller pieces, and digested overnight at 4°C in HBSS without $Ca^{2+}$ and $Mg^{2+}$ (Thermo Fisher Scientific), containing 1 ml dispase (5,000 caseinolytic units/ml; Corning) for every 9 ml of HBSS containing final concentrations of 100 mg/ml streptomycin, 100 U/ml penicillin, and 250 ng/ml amphotericin B (HyClone). After digestion, the epidermis was removed and discarded, whereas the remaining dermis was cut into smaller pieces less than a few square millimeter in area. These pieces were moistened with DMEM and pressed into a six-well plate scored with a razor blade. The dermis was maintained in DMEM containing 10% FBS and 1% vol/vol penicillin/streptomycin solution at 37°C, 5% $CO_2$. Media was changed every 4–5 d and fibroblast growth was typically observed within 1 wk of culture. Once sufficient outgrowth had occurred, the dermis was removed from the plate and the fibroblasts expanded into larger cultures. Complete donor information as provided by these sources can be found in Table 4.

### Cell culture

All cells, unless otherwise stated, were grown under standard conditions in DMEM (Thermo Fisher Scientific) containing 10% FBS and 1% vol/vol penicillin/streptomycin. Upon reaching confluency, the cells were trypsinized with 0.05% trypsin/EDTA and split 1:3.

### polyI:C transfections

DFs were transfected with ~0.05 μg of high molecular weight poly(I:C) (Invivogen) per square centimeter (0.5 μg/well in a six-well format in triplicate for samples undergoing RNA-Seq and 0.1 μg/well in duplicate in a 24-well format for relative RT-qPCR validation experiments) using X-tremeGENE HP DNA Transfection Reagent (Roche) 1 μl per μg of poly(I:C) in Opti-MEM (Thermo Fisher Scientific). Mock transfections were performed in parallel under the same conditions minus poly(I:C). Collected cell lysates (350 μl volume) were immediately frozen at −80°C until RNA extraction performed.

### RNA extraction, cDNA library preparation, and RNA sequencing

Total RNA was isolated from DFs using the RNeasy Mini Kit (QIAGEN) according to the manufacturer's instructions. For all samples undergoing RNA-Seq, the quality and concentration of RNA was assessed on an Agilent 2100 Bioanalyzer (Agilent Technologies). All samples subsequently sequenced had an RNA Integrity Number of ≥8. The poly-A–containing RNA transcripts in the total RNA samples were converted to cDNA and amplified after the Smart-seq2 method (49). Sequencing libraries were made from the amplified cDNA samples using the Nextera kit (Illumina), assigning a unique barcode

to each of the libraries to be sequenced together. The cDNA samples and libraries were examined on the Bioanalyzer (Agilent) DNA HS chips for size distribution and quantified by Qubit fluorometer (Invitrogen). The RNA-Seq libraries were pooled together in equal amounts and sequenced on the Illumina HiSeq 2500 Rapid Flowcell as single-end 75-nt reads following the standard protocol, giving a range of 15–20 million reads per sample. Raw sequencing reads were filtered by Illumina HiSeq Control Software and only pass-filter reads were used for further analysis.

### RT–qPCR of select ISGs

RT–qPCR of total RNA isolated from the 24-well format poly(I:C) transfections was performed using the Luna Universal One-Step RT-qPCR kit (New England BioLabs, Inc.) according to the manufacturer's directions. Primer sequences can be found in Table 6. In brief, a master

mix was prepared that comprised 10 $\mu$l of 2× Luna Universal One-Step Reaction Mix (2×), 1 $\mu$l of 20× Luna WarmStart RT Enzyme Mix, 0.8 $\mu$l of a 10 $\mu$M stock of each primer, and 5.4 $\mu$l of nuclease-free water per reaction. Each well received 18 $\mu$l of the appropriate master mix and 2 $\mu$l of the RNA being assayed. The following PCR program was then run on an Applied Biosystems Step One Plus qPCR machine (Life Technologies): denatured at 55°C for 10 min, 95°C for 1 min, followed by 40 cycles of 95°C for 10 s and 60°C for 60 s. Last, a melt curve was performed at 95°C for 10 s, 65°C for 10 s, 95°C for 10 s, and 50°C for 5 s.

### RNA-Seq analysis

Using the Galaxy system (50) provided by Princeton University, short reads were aligned to the human or the originating species' genomes (see Table 2) using RNA STAR (51, 52) (Galaxy version 2.6.0b-1)

**Table 6.  Primers for RT-qPCR of IFNβ, MX1, OASL, and HPRT1 for the species used in this study.**

| Species | Mx1 | OasL (Oasl1 for mouse) | HPRT1 | IFNB |
|---|---|---|---|---|
| Rhesus | PU-O-2198, -2532 | PU-O-4828, -4829 | PU-O-2409, -1469 | PU-O-2211, -2215 |
| Bonobo | PU-O-2195, -2196 | PU-O-2206, -2209 | PU-O-1468, -1469 | PU-O-2211, -2212 |
| Chimpanzee | PU-O-2195, -2196 | PU-O-2206, -2209 | PU-O-1468, -1469 | PU-O-2211, -2212 |
| Pigtailed macaque | PU-O-2198, -2196 | PU-O-4828, -4829 | PU-O-1468, -1469 | PU-O-2211, -2215 |
| Squirrel monkey | PU-O-2525, -2526 | PU-O-2529, -2530 | PU-O-1468, -1469 | PU-O-2475, -2476 |
| Orangutan | PU-O-2197, -2199 | PU-O-2206, -2208 | PU-O-1468, -1469 | PU-O-2211, -2214 |
| Mouse | PU-O-4236, -4237 | PU-O-4856, -4857 | PU-O-4212, -4213 | PU-O-4240, -4241 |
| Olive baboon | PU-O-2196, -2198 | PU-O-2207, -2210 | PU-O-1468, -1469 | PU-O-2211, -2215 |
| Gorilla | PU-O-2195, -2196 | PU-O-2206, -2208 | PU-O-1468, -1469 | PU-O-2211, -2216 |
| Human | PU-O-2195, -2196 | PU-O-2206, -2208 | PU-O-1468, -1469 | PU-O-2211, -2212 |
| Forward primers (5′ to 3′) | | Reverse primers (5′ to 3′) | | |
| PU-O-1468 | CCTGGCGTCGTGATTAGTGAT | PU-O-1469 | AGACGTTCAGTCCTGTCCATAA | |
| PU-O-2195 | GTTTCCGAAGTGGACATCGCA | PU-O-2196 | CTGCACAGGTTGTTCTCAGC | |
| PU-O-2197 | CTTTCCGAAGTGGACATCGCA | PU-O-2199 | CTGCACAGATTGTTCTCAGC | |
| PU-O-2198 | CTTTCTGAAGTGGACATTGTA | PU-O-2208 | CACAGCGTCTAGCACCTCTT | |
| PU-O-2206 | CTGATGCAGGAACTGTATAGC | PU-O-2209 | CACAGTGTCTAGCACCTCTT | |
| PU-O-2207 | CTGATGCAGGAACTGTACAGC | PU-O-2210 | CACAGCATCTAGAACCTCCT | |
| PU-O-2211 | GCTTGGATTCCTACAAAGAAGCA | PU-O-2212 | ATAGATGGTCAATGCGGCGTC | |
| PU-O-2409 | GATTAGTGATGATGAACCA | PU-O-2214 | GTAGATGGTCAATGCCGCGTC | |
| PU-O-2475 | ACTTGGATTCCTACAAAGAAGAA | PU-O-2215 | ATAGATGGTCAATGCAGCGTC | |
| PU-O-2525 | CTTTCCGAAGTGGGAGTCGGA | PU-O-2216 | ATAGATGGTCAATGCCGCGTC | |
| PU-O-2529 | CTGACACAGGAGCTGTATGCC | PU-O-2476 | ATAGACGATTAATGCCACGTC | |
| PU-O-4212 | TCAGTCAACGGGGGACATAAA | PU-O-2526 | CTGTACAGGTTGTTCTCGGC | |
| PU-O-4236 | GGTCTTGGATGTGATGCGGA | PU-O-2530 | CACAGTGTCCAGCACCTCTT | |
| PU-O-4240 | TGTCCTCAACTGCTCTCCAC | PU-O-2532 | TGCACAGGTTGTTCTCAGC | |
| PU-O-4828 | CCATCGTGCCTGCCTACAGAG | PU-O-4213 | GGGGCTGTACTGCTTAACCAG | |
| PU-O-4856 | CAGGAGCTGTACGGCTTCC | PU-O-4237 | TGCTGACCTCTGCACTTGAC | |
| | | PU-O-4241 | ACCACCACTCATTCTGAGGC | |
| | | PU-O-4829 | CTTCAGCTTAGTTGGCCGATG | |
| | | PU-O-4857 | CCTACCTTGAGTACCTTGAGCAC | |

with default parameters. Counts were generated using featureCounts with default settings (53) (Galaxy version 1.6.3+galaxy2), downloaded, and read into R (54) version 3.5.2 (December 20, 2018) using scripts run in RStudio (55) version 1.1.463. DGE was determined using DESeq2 (56) (version 1.22.2) using the standard filters and nbinomWaldTest (57). The design used to model the samples was ~species + species: donor.n + treatment + species:treatment with various contrasts set as shown in the R code and as described in the README files for the Datasets EV3–5. Transcript counts were normalized using DESeq2 default options and transformed using the regularized $\log_2$ function in DESeq2 before PCA plotting. Results were extracted from the DESeq2 analysis and annotated using Bioconductor's AnnotationDbi (version 1.44.0) org.Hs.eg.db and org.Mm.eg.db. For further details concerning the R packages used and the specific conditions used for analysis, R code can be accessed at https://github.com/aploss/polyIC-dermal-fibroblasts-RNA-Seq.

## Statistical analysis

Statistical analysis for RT-qPCR data was performed with GraphPad Prism software as indicated in the figure legends.

## Data access

The RNA-Seq data generated in this study are deposited in the National Center for Biotechnology Information Gene Expression Omnibus database (accession number GSE105160).

# Supplementary Information

# Acknowledgements

This work was supported by grants from the National Institutes of Health (R01 AI107301 to A Ploss, R21AI117213 to A Ploss and RE Schwartz), a Research Scholar Award from the American Cancer Society (RSG-15-048-01-MPC to A Ploss), Princeton University, a research grant through the Princeton University Center for Health and Wellbeing Health Grand Challenge Program to A Ploss and JM Gaska, and an Investigator in Pathogenesis Award by the Burroughs Wellcome Fund (101539) to A Ploss. JM Gaska was supported by the National Institute of General Medical Sciences of the National Institutes of Health under grant number T32GM007388. We also thank Jennifer Miller and Jessica Wiggins of the Lewis-Sigler Institute for Integrative Genomics for their help in assessing the RNA quality, preparing the cDNA libraries, and performing RNA-Seq. We thank members of the Ploss lab for critical discussions and comments on this project. We thank Matthew Cahn (Princeton University) for his assistance in facilitating the submission of our raw and processed data to National Center for Biotechnology Information Gene Expression Omnibus (accession number GSE105160).

## Author Contributions

JM Gaska: conceptualization, formal analysis, funding acquisition, investigation, visualization, methodology, and writing—original draft and editing.

L Parsons: formal analysis, methodology, and writing—review and editing.
M Balev: formal analysis.
A Cirincione: formal analysis.
W Wang: formal analysis, investigation, and methodology.
RE Schwartz: conceptualization.
A Ploss: conceptualization, supervision, funding acquisition, project administration, and writing—original draft, review, and editing.

## Conflict of Interest Statement

The authors declare that they have no conflict of interest.

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
