## [Reviewer comments · Life Science Alliance]

Conservation of cell-intrinsic immune responses in diverse non-human primate species

Jenna M. Gaska, Lance Parsons, Metodi Balev, Ann Cirincione, Wei Wang, Robert E. Schwartz, Alexander Ploss

DOI: 10.26508/lsa.201900495

Corresponding author(s): Prof. Alexander Ploss (Princeton University)

Review timeline:

Submission Date:	2019-07-20
Editorial Decision:	2019-08-13
Revision Received:	2019-09-18
Editorial Decision:	2019-10-09
Revision Received:	2019-10-14
Accepted:	2019-10-15

Scientific Editor: Andrea Leibfried

Transaction Report:

No Peer Review Process File is available with this article, as the authors have chosen not to make the review process public in this case.

1st Editorial Decision

13 August 2019

August 13, 2019

Re: Life Science Alliance manuscript #LSA-2019-00495-T

Alexander Ploss
Princeton University
Department of Molecular Biology
Washington Road
Lewis Thomas Laboratory 110
Princeton, NJ 8544

Dear Dr. Ploss,

Thank you for submitting your manuscript entitled "Conservation of cell-intrinsic immune responses in diverse non-human primate species" to Life Science Alliance. The manuscript was assessed by expert reviewers, whose comments are appended to this letter.

As you will see, the reviewers acknowledge that there is interest in your work, but question at this stage the significance of your findings and the usefulness of the data generated. They provide, however, constructive input on how to address these issues (and partially by using the data already at hand). We would therefore like to invite you to submit a revised version of your manuscript, addressing all reviewers concerns and, importantly, those noted above. Should this prove too difficult or should you want to discuss an individual revision point further with us, please do get in touch.

Thank you for this interesting contribution to Life Science Alliance. We are looking forward to receiving your revised manuscript.

Sincerely,

B. MANUSCRIPT ORGANIZATION AND FORMATTING:

2nd Editorial Decision

09 October 2019

October 9, 2019

RE: Life Science Alliance Manuscript #LSA-2019-00495-TR

Prof. Alexander Ploss
Princeton University
Department of Molecular Biology
Washington Road
Lewis Thomas Laboratory 110
Princeton, NJ 8544

Dear Dr. Ploss,

Thank you for submitting your revised manuscript entitled "Conservation of cell-intrinsic immune responses in diverse non-human primate species". As you will see, while reviewer #2 still thinks that the work would be much stronger if increasing power and validation, both reviewers appreciate your responses and the introduced changes. We would thus be happy to publish your paper in Life Science Alliance pending final revisions, mainly necessary to meet our formatting guidelines:

- please incorporate the text change suggested by rev#2
- please provide in the word doc manuscript file titles/short legends for the supplementary tables and for the datasets

A. FINAL FILES:

B. MANUSCRIPT ORGANIZATION AND FORMATTING:

Sincerely,

Andrea Leibfried, PhD
Executive Editor
Life Science Alliance
Meyershofstr. 1
69117 Heidelberg, Germany
t +49 6221 8891 502
e a.leibfried@life-science-alliance.org
www.life-science-alliance.org

3rd Editorial Decision

15 October 2019

RE: Life Science Alliance Manuscript #LSA-2019-00495-TRR

Prof. Alexander Ploss
Princeton University
Department of Molecular Biology
Washington Road
Lewis Thomas Laboratory 110
Princeton, NJ 8544

Dear Dr. Ploss,

Thank you for submitting your Resource entitled "Conservation of cell-intrinsic immune responses in diverse non-human primate species". It is a pleasure to let you know that your manuscript is now accepted for publication in Life Science Alliance. Congratulations on this interesting work.

DISTRIBUTION OF MATERIALS:

Again, congratulations on a very nice paper. I hope you found the review process to be constructive and are pleased with how the manuscript was handled editorially. We look forward to future exciting submissions from your lab.

Sincerely,

Andrea Leibfried, PhD

Executive Editor
Life Science Alliance
Meyerhofstr. 1
69117 Heidelberg, Germany
t +49 6221 8891 502
e a.leibfried@life-science-alliance.org
www.life-science-alliance.org